# Long Intergenic Non-Protein Coding RNA 173 in Human Cancers

**DOI:** 10.3390/cancers14235923

**Published:** 2022-11-30

**Authors:** Wei Mao, Yi Liao, Liling Tang

**Affiliations:** 1Key Laboratory of Biorheological Science and Technology, Ministry of Education, College of Bioengineering, Chongqing University, Chongqing 400044, China; 2Department of Thoracic Surgery, Southwest Hospital, Army Medical University (Third Military Medical University), Chongqing 400038, China

**Keywords:** LINC00173, cancer, target, chemoresistance, ceRNAs

## Abstract

**Simple Summary:**

Numerous studies have shown that long non-coding RNAs (lncRNAs) can regulate the development and progression of human cancers in various ways. Long intergenic non-coding RNA 173 (LINC00173), as a recently discovered non-coding gene, is dysregulated in 11 human cancers. Its aberrant expression is closely associated with diversified biological processes such as proliferation, migration, and invasion. This review systematically summarizes the latest findings concerning the expression and the regulatory mechanisms of LINC00173, which covers the tumorigenesis and chemoresistance, and regulatory targets of LINC00173, thus providing a novel biomarker for cancer prognosis and a therapeutic target.

**Abstract:**

Long non-coding RNAs belong to non-coding RNAs (ncRNAs) with a length of more than 200 nucleotides and limited protein-coding ability. Growing research has clarified that dysregulated lncRNAs are correlated with the development of various complex diseases, including cancer. LINC00173 has drawn researchers’ attention as one of the recently discovered lncRNAs. Aberrant expression of LINC00173 affects the initiation and progression of human cancers. In the present review, we summarize the recent considerable research on LINC00173 in 11 human cancers. Through the summary of the abnormal expression of LINC00173 and its potential molecular regulation mechanism in cancers, this article indicates that LINC00173 may serve as a potential diagnostic biomarker and a target for drug therapy, thus providing novel clues for future related research.

## 1. Introduction

In recent years, the advancement of high-throughput techniques has allowed in-depth research of the human non-coding genomic region [1]. Impressively, the vast majority of RNAs transcribed from the human genome, except mRNA, are ncRNAs, which exert influences on normal gene expression and disease development [2]. There are two categories of ncRNAs based on their length: lncRNAs and small ncRNAs [3]. LncRNAs, a kind of ncRNA with a transcript of more than 200 nucleotides, were once thought to be transcriptional “noise” because of their inability to code protein [4]. Long intergenic non-coding RNAs (lincRNAs) are a particular kind of lncRNA that do not overlap with any protein-coding locus [5]. LincRNAs have been proven to be aberrantly expressed in various malignant tumors and participate in the progression of human diseases. For example, LINC00337 served as an oncogenic lincRNA that aggravated the progression of lung adenocarcinoma [6]. LINC01977 could facilitate breast cancer (BC) development and chemoresistance to doxorubicin [7]. LincRNA-p21 could regulate neuroinflammation and autophagy in Alzheimer’s disease [8].

LINC00173, also known as NCRNA00173, is a not fully characterized lncRNA located on chromosome 12q24.22 [9], with ~543 nucleotides and no protein-coding potential [10]. There are four transcript variants for LINC00173, of which TSV1 and TSV2 have been the most studied [11]. Subcellular fractionation analysis shows that LINC00173 is located both in the cytoplasm and nucleus [12]. However, the proportion of its subcellular localization differs among various cancer types. It is well known that the function of lncRNAs depends greatly on their unique subcellular localization, implying that the interacting factors of lncRNAs and the mechanism on which lncRNAs exert their functions are largely related to their localization [13]. Nuclear lncRNAs are essential for several biological processes, such as chromatin organization and transcriptional gene expression [14]. In contrast, cytoplasmic lncRNAs are mainly responsible for post-transcriptional regulation [15]. LINC00173 exerts biological functions both in transcription and post-transcription.

LINC00173 was first verified as a new regulatory factor in the development of granulocyte [9]. Soon afterward, LINC00173 was initially identified in cells infected with HIV-1 [16] and was first systematically studied in lung cancer (LC) [17]. The abnormal expression and regulatory function of LINC00173 were verified in several tumors, including BC [7], esophageal squamous cell carcinoma (ESCC) [18], and gastric carcinoma (GC) [19]. On the other hand, LINC00173 has been proven to serve as a vital regulator of several non-tumor diseases, such as hypertrophic scar fibroblasts [20] and polycystic ovarian syndrome (PCOS) [21], regulating a series of basic life activities, including cell proliferation, differentiation, and apoptosis [22].

Therefore, as a new type of lncRNA, LINC0173 is vital for human cancer research. However, its biological function and mechanism in the development of cancers are still being explored. In this paper, we summarized the roles of LINC00173 in some different cancers and the molecular mechanism, including tumorigenesis, chemoresistance, and its regulatory targets.

## 2. Aberrant Expression and Tumorigenesis of LINC00173 in Human Cancer

LncRNAs are frequently aberrant expressions in malignancies and are connected with further cancer-related genetic modifications [23]. Furthermore, dysregulated expression lncRNAs are inextricably linked to tumorigenesis [24]. As shown in Table 1, the high expression level of LINC00173 has been verified in several human cancers, including LC [17,25,26,27,28], hepatocellular carcinoma (HCC) [29], colorectal cancer (CRC) [30], ESCC [18], pancreatic cancer (PCA) [11], glioma [31], triple-negative breast cancer (TNBC) [32], prostate cancer (Pca) [33], and WT [34]. In contrast, LINC00173 is downregulated in cervical cancer (CC) [12], acute myeloid leukemia (AML) [35], and non-small cell lung cancer (NSCLC) [17]. Additionally, LINC00173 performs the role of oncogenic lncRNA or tumor suppressor in cancers. Meanwhile, dysregulated LINC00173 is involved in several physiological and pathological processes of human cancers with multiple functions and mechanisms. This section focuses on the differential expression and the regulatory function of LINC00173 in a variety of tumors.

### 2.1. Oncogenic

#### 2.1.1. Lung Cancer

LC is not only the second most common cancer worldwide but also the main cause of mortality from cancer [36]. Histopathologically, LC is divided into two categories: NSCLC and small cell lung cancer (SCLC) [37]. The 5-year survival time of LC still ranges between 4 and 17% despite the discovery of chemotherapy, radiotherapy, and epidermal growth factor receptor-directed treatments [38]. Meanwhile, the prognosis of LC patients remains poor [39]. Thus, it is extremely necessary to exploit more reliable screening and diagnostic markers. Nearly 85% of LC cases are confirmed to be NSCLC depending on the tissue subtype, and its occurrence rate is increasing continuously worldwide [40]. Yang et al. [25] found that the serum expression level of LINC00173 in NSCLC patients, compared with healthy patients and patients with benign pulmonary disease, was higher. This result suggested that LINC00173 could function as a useful non-invasive diagnosis biomarker for NSCLC patients. Meanwhile, Chen et al. [26] showed that the expression of LINC00173.v1 and LINC00173.v2 were both highly expressed in lung squamous cell carcinoma (SQC) cells, and LINC00173.v1 was predominantly expressed in LC tissues and especially upregulated in SQC cells. In addition, LINC00173.v1 overexpression promoted the metastasis and early relapse of SQC patients, while the silencing of LINC00173.v1 had a negative effect on the proliferation and migration ability of vascular endothelial cells and the angiogenesis of LC cells. Taken together, their study may contribute to the research on new antiangiogenic strategies. Zeng et al. [28] revealed that LINC00173 was highly increased in chemoresistant SCLC cell lines and was closely correlated with chemoresistance, extensive stage, and shorter survival time. In addition, LINC00173 enhanced the chemoresistance, migration, growth, and invasion of SCLC in vitro, and it enhanced the chemoresistance and proliferation of SCLC cells in vivo as well. Wang et al. [41] also showed that LINC00173 was upregulated in chemoresistant SCLC cell lines and could enhance the chemoresistance and development of SCLC cells by affecting glucose metabolism.

Taken together, the aforementioned results suggest that LINC00173 may be a promising diagnostic biomarker and novel therapeutic target for the treatment of LC.

#### 2.1.2. Hepatocellular Carcinoma

HCC is one of the most commonly diagnosed cancers in Asia [42]. LINC00173 has been verified to be associated with digestive system cancer. Zhao et al. [29] reported that LINC00173 was notably upregulated in cisplatin (CDDP)-resistant HCC tissues and cell lines. A high expression level of LINC00173 was associated with a shorter overall survival time in HCC patients. Functionally, LINC00173 silencing enhanced the sensitivity of CDDP-resistant HCC cells to CDDP. Mechanistically, LINC00173 sponged miR-641 to improve CDDP resistance in HCC patients, which could offer new clues for therapeutic strategies for HCC.

#### 2.1.3. Colorectal Cancer

CRC is the third leading cause of cancer mortality all over the world, and understanding the incidence of CRC is significant in order to exploit more molecular targeting [43]. Yu et al. [30] found that LINC00173 was elevated in 57 pairs of CRC tissues and cell lines. LINC00173 knockdown hindered the migration, invasion, growth, and chemoresistance of CRC cells. In addition, the xenograft assay showed that LINC00173 knockdown extremely repressed tumor growth in vivo.

#### 2.1.4. Esophageal Squamous Cell Carcinoma

ESCC as the main cause of cancer-associated death has a poor prognosis worldwide [44]. Mao et al. [18] noted that LINC00173 from a seven-lncRNA signature was upregulated, and knockdown of LINC00173 could increase the colony numbers and decrease the G1/G0 population of ESCC cells. Thus, LINC00173 may play a critical role in ESCC survival prediction.

#### 2.1.5. Glioma

Recent studies have pointed out that lncRNAs are dysregulated in the growth and development of the nervous system [45]. As one of the most common primary central nervous system malignancies, glioma has a high degree of heterogeneity and poor prognosis [46]. Du et al. [31] showed that LINC00173 was upregulated in glioma tissues and cells. Then, the depletion of LINC00173 decreased the proliferation, migration, and invasion of glioma cells.

#### 2.1.6. Breast Cancer

BC is one of the most commonly diagnosed and deadly cancers among women. TNBC, which is characterized by extreme aggressiveness, is a subtype of BC [47]. Fan et al. [32] found that the expression level of LINC00173 was markedly increased relative to paired normal tissues. Both loss and gain-of-function methods have proven that LINC00173 plays an oncogenic role in promoting TNBC cell proliferation and invasion in vitro. Additionally, silencing of LINC00173 inhibited MDAMB-231 cell-induced tumorigenesis in vivo.

#### 2.1.7. Prostate Cancer

PCa, a malignant epithelial tumor, has a high incidence in men [48]. Hu et al. [33] pointed out that LINC00173 was elevated in PCa patients, and its expression level was positively correlated with cancer stage and metastasis. In addition, upregulated LINC00173 was clinically linked to poor prognosis in PCa patients. Additionally, LINC00173 played a role in the metastasis and proliferation of PCa cells. Their data imply that LINC00173 could be a new biomarker and potential therapeutic target for PCa.

#### 2.1.8. Wilms’ Tumor

The kidney is in charge of the urinary system, which serves as the human body’s chief organ for filtering blood and eliminating metabolic waste from the body through the glomerulus [49]. WT is a kind of kidney cancer that occurs in childhood, and approximately one in 10,000 children suffer from WT [50]. Zhu et al. [34] recently reported that LINC00173 was upregulated in WT cell lines more than that in the non-tumor human embryonic kidney cell lines. Furthermore, LINC00173 knockdown attenuated WT cell migration and invasion and enhanced apoptosis. Additionally, LINC00173 depletion restrained WT tumor growth and metastasis in vivo.

### 2.2. Tumor Suppressor

#### 2.2.1. Non-small Cell Lung Cancer

LINC00173 also exerts a tumor suppressor function in NSCLC. Yang et al. [17] showed that the expression level of LINC00173 in NSCLC tissues was dramatically lower than that in adjacent normal tissues. Knockdown of LINC00173 increased the proliferation, migration, and apoptosis rate of NSCLC cells, while overexpression of LINC00173 exhibited the opposite cell behaviors.

#### 2.2.2. Pancreatic Cancer

PCA is a deadly disease with a high recurrence rate and poor prognosis in the early stages, even in the context of resectable pancreatic cancer [51]. Liu et al. [11] found that TSV1 and TSV2 of LINC00173 were both highly upregulated in PCA cell lines and tissues. Biological functional studies in vitro have shown that overexpression of LINC00173 increases cancer cell apoptosis while inhibiting PCA cell invasion and proliferation. Moreover, xenograft mouse models have shown that LINC00173 overexpression attenuates PCA growth in vivo. The aforementioned findings suggest that LINC00173 functions as a tumor suppressor lncRNA and contributes to PCA development [11].

#### 2.2.3. Cervical Cancer

CC is one of the most commonly diagnosed female malignant tumors, and a high relapse rate is often coupled with the treatment of CC [52]. Zhang et al. [12] showed that LINC00173 was downregulated in CC tissues compared with that in healthy tissues. LINC00173 knockdown inhibited the proliferation, invasion, migration, and 5-FU resistance of CC cells and significantly decreased tumor growth in vivo.

#### 2.2.4. Acute Myeloid Leukemia

AML is a malignant disease characterized by highly heterogeneous and aggressive hematological and poor treatment outcomes [53]. AML can be caused by long-term exposure to benzene or hydroquinone (HQ). Zhang et al. [35] found that LINC0073 was downregulated in HQ-MT, benzene-exposed workers, and leukemia cell lines. LINC00173 overexpression repressed the proliferation and enhanced the apoptosis and sensitivity to CDDP of HQ-MT cells. In addition, the overexpression of LINC00173 attenuated tumorigenicity in xenograft tumor models. Additionally, Schwarzer et al. found that LINC00173 played a crucial role in blood hierarchy formation and maintenance [9].

## 3. Biological Roles of LINC00173

### 3.1. LINC00173 and Chemoresistance

Chemotherapy is currently the preferred choice in the treatment of human cancers [54]. However, The primary factor in cancer chemotherapy treatment failure is drug resistance [55]. Recent efforts have revealed that lncRNAs exhibit their regulatory functions in cancer biology, including chemoresistance [56]. LncRNAs are closely related to cancer drug resistance [57]. As shown in Figure 1, LINC00173 can modulate human cancer chemotherapy sensitivity through different mechanisms. This part will briefly depict the role of LINC00173 in the chemoresistance of SQC [26], SCLC [28,41], HCC [29], CRC [30], and AML [35].

CDDP is a commonly used cytotoxic antitumor drug that can enhance DNA damage and alter the cellular metabolic state of the tumor cell to cause apoptosis [58]. In HCC [29], SQC [30], and AML caused by benzene or HQ [35], LINC00173 can enhance the CDDP resistance of those cancer cells. In HCC, LINC00173 upregulated RAB14 to enhance the CDDP resistance of HCC cells by targeting miR-641. This may provide a potential therapeutic target for HCC [29]. 5-Aza-2′-deoxycytidine (5-AzaC) can increase the CDDP sensitivity of cancer cells [59]. After 5-AzaC treatment, LINC00173 was decreased in HQ-MT cells, suggesting that LINC00173 is related to CDDP resistance in HQ-MT cells. Subsequently, the assessment of inhibitory concentration 50% (IC50) value and cancer cell viability shows that LINC00173 overexpression can promote the CDDP resistance of HQ-MT cells. Additionally, an SQC in vivo xenograft mouse model demonstrated that LINC00173.v1 ASO repressed tumor growth and also dramatically promoted the therapeutic sensitivity of SQC cells against CDDP, with efficiency equivalent to even the combined effect of bevacizumab and CDDP. This indicated that LINC00173.v1 can serve as a chemotherapeutic sensitizer in SQC [30].

Collectively, the aforementioned studies indicate that LINC00173 may be a new target for treating CDDP resistance.

As a fluoropyrimidine drug, 5-fluorouracil (5-FU) has certain cytotoxicity and is a common treatment drug for several solid malignancies [60]. To investigate the effect of LINC00173 on drug resistance in CRC, researchers treated CRC cells with 5-FU after transfection, and the results showed that LINC00173 promoted the 5-FU resistance of CRC cells by binding the miR-765/PLP2 axis [30].

There are two ways that LINC00173 regulates multidrug resistance in SCLC: one is that LINC00173 performs a chemoresistance-promoting function by regulating glucose metabolism [41], and the other is that LINC00173 promotes the chemoresistance of SCLC by binding with miR-218 to regulate the expression of Etk [28].

In summary, LINC00173 can serve as a potential drug target for chemotherapy in multiple human cancers.

### 3.2. Regulatory Targets of LINC00173

#### 3.2.1. LINC00173 and Competing Endogenous RNAs (ceRNAs)

MicroRNAs (miRNAs) are short non-coding RNAs, usually consisting of 19–25 nucleotides, that specifically bind to the downstream target mRNAs to hinder gene degradation or translation [61]. Meanwhile, miRNAs can regulate the occurrence and development of human cancers by acting as oncogenes or tumor suppressor genes [62]. Multiple studies have demonstrated that lncRNAs affect tumor progression in multiple ways, such as serving as ceRNAs for miRNAs binding, thereby restoring the expression ability and activity of downstream genes [63]. Recently, the mechanism of lncRNAs, as ceRNAs of miRNAs to eliminate inhibition of targeted genes mediated by miRNA, has attracted much attention [64]. CeRNAs regulatory networks involving lncRNAs play an important role in human cancers. As shown in Figure 2, LINC00173, as a ceRNA, directly or indirectly participates in regulating the biological processes of cancer diseases, and involves eight miRNAs: miR-182-5p [12,17], miR-511-5p [26], miR-218 [28], miR-641 [29], miR-765 [30,31], miR-490-3p [32], miR-124-3p [21], and miR-338-3p [33].

First, LINC00173 is involved in the development of LC by binding with the different miRNAs. In SCLC, LINC00173 promotes the proliferation, migration, invasion, chemoresistance, and tumorigenesis of cancer cells by targeting miR-218 to modulate Etk expression [28]. ΔNp63α, an SQC-specific factor, causes LINC00173.v1 overexpression by regulating the transcriptional response of SQC cells. Upregulated LINC00173.v1 further promotes the migration and proliferation of endothelial cells and inhibits miR-511-5p expression by sponging miR-511-5p. This finding suggests that the miR-511-5p/VEGFA axis plays a vital role in LINC00173.v1-induced tumorigenesis and could serve as a target for antiangiogenic therapeutic therapy in SQC [26].

Second, LINC00173 contributes to the development of different tumors by binding with the same miRNA. In glioma, LINC00173 sponges miR-765 to upregulate the expression of NUTF2 expression, and NUTF2 overexpression further promotes the proliferation, migration, and invasion of glioma cells, suggesting that LINC00173 functions as a critical oncogenic lncRNA in glioma [31]. In CRC, LINC00173 promotes CRC cell proliferation, invasion, and chemoresistance by interacting with miR-765 to elevate PLP2 expression [30]. Taken together, LINC00173 interacts with the same miRNA, miR-765, to exert its regulatory function in both glioma and CRC.

Finally, LINC00173 is downregulated in both CC [12] and NSCLC [17] tissues, and it competitively induces miR-182-5p accumulation. In contrast, LINC00173 exerts a suppressive function in CC but a carcinogenic role in NSCLC. FBXW7 is a tumor suppressor gene and inhibits the proliferation and invasiveness of CC cells [65]. In CC, diminished LINC00173 acts as an miRNA to upregulate miR-182-5p and in turn inhibits the proliferation, migration, and invasion of CC tumor cells by upregulating FBXW7 [12]. Abnormal AGER expression is attributed to tumorigenesis [66]. In NSCLC, downregulated LINC00173 negatively regulates miR-182-5p, promotes proliferation and migration, and inhibits apoptosis of NSCLC cells via the miR-182-5p/AGER/NF-κB pathway. In HCC, upregulated LINC00173 enhances the CDDP resistance of HCC cells through the miR-641/RAB14 axis [29]. In PCa, LINC00173 sponges miR-338-3p and promotes PCa cell proliferation, migration, metastasis, and invasion by upregulating Rab25 [33]. In TNBC [32], LINC00173 increases TNBC cells’ proliferative and invasive capabilities by inhibiting the expression of miR-490-3p [32].

#### 3.2.2. Other Targets of LINC00173

DNA methylation, an epigenetic modification, plays a vital role in regulating gene expression [59]. DNA methyltransferases (DNMTs) can typically add a methyl group to the cytosine in the promoter region of the target gene, thereby blocking transcription, and resulting in the downregulation of the target gene [67]. In AML, DNA methyltransferase 1 (DNMT1), a maintainer of DNA methylation during DNA replication [68], is upregulated and decreases LINC00173 expression by promoting methylation of LINC00173, thereby modulating HQ-MT cell proliferation and apoptosis [35].

Y-box binding protein 1 (YB1), known as a protein member of the cold shock domain (CSD) protein family, can promote the progression of multiple cancers and acts as a possible biomarker and therapeutic target [69]. In SCLC, LINC00173 directly interacts with the YB1 protein, further promoting the nuclear translocation and phosphorylation of YB1. In addition, the upregulated YB1 protein increases the expression of the glucose-6-phosphate dehydrogenase (G6PD) and glucose metabolic enzymes hexokinase 2 (HK2), which respectively activates the pentose phosphate pathway (PPP) and glycolysis. This research may provide a potential therapeutic method for SCLC treatment by targeting the PPP and glycolysis.

SPHK1 is a rate-limiting enzyme for the synthesis of sphingosine 1 phosphate and plays a key role in tumorigenesis. Additionally, SPHK1 promotes cell proliferation and apoptosis in a manner dependent on the Akt/NF-kB pathway [70]. In PCA, high expression of LINC00173 inhibits SPHK1 expression, which then hinders activated p-Akt and NF-kB p65 expression in MIA PaCa-2 cells [11].

MGAT1 is an N-acetylglucosaminyltransferase I (GlcNAcT-1) that initiates the process of complex N-glycans synthesis [71]. Previous studies have elucidated that RNA-binding proteins (RBPs) are a key mediator in the biological processes of the post-transcriptional regulation of genes by lncRNAs [72]. In WT cells, overexpression of LINC00173 stabilized MGAT1 mRNA via recruiting HNRNPA2B1 to upregulate MGAT1 expression. Moreover, MGAT1 repressed apoptosis and enhanced cell migration by stabilizing the expression of the MUC3A protein via N-glycosylation in WT cells. Taken together, their findings could provide support for exploring novel therapeutic targets for WT patients [34].

## 4. Conclusions and Perspectives

An increasing number of studies have verified that LINC00173 is dysregulated in a variety of tumor tissues and cancer cell lines, suggesting that LINC00173 plays a key role in the development and progression of multiple human cancers. LINC00173 regulates various cellular biological processes, including proliferation, apoptosis, invasion, migration, and chemotherapeutic sensitivity. Mechanistically, LINC00173 functions as the specific sponge of miRNAs or ceRNAs to regulate the downstream target genes’ expression. In addition, the differential expression of LINC00173 in several cancers makes it a potential biomarker for diagnosis and prognosis.

Currently, the different functional roles of LINC00173 in LC, HCC, CRC, PCA, ESCC, glioma, BC, PCa, WT, and AML have been studied. However, studies have not fully revealed the functions of LINC00173 in other cancers. On the other hand, five reports predict that LINC00173 may be involved in the development of diseases through bioinformatics analysis [19,73,74,75,76], Thus, it is necessary to study the biological function and molecular mechanism of LINC00173 in human diseases by molecular experiments.

In conclusion, LINC00173, as a novel lncRNA, has only recently been extensively studied. Its biological characteristics in human cancer diagnosis, treatment, and prognosis assessment are still poorly understood. Thus, the role of LINC00173 in human cancers needs to be further investigated. With the deepening of research, LINC00173 is expected to become a promising biomarker for and therapeutic target in human diseases.

## Figures and Tables

**Figure 1 cancers-14-05923-f001:**
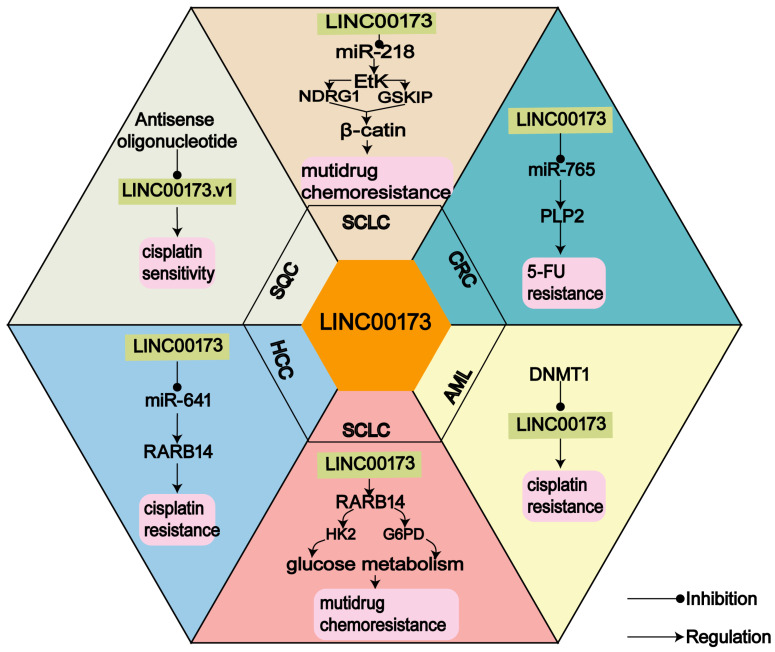
The biological function of LINC00173 in chemoresistance. LINC00173 regulates drug resistance in 6 human cancers, including AML, SCLC, HCC, CRC, and SQC.

**Figure 2 cancers-14-05923-f002:**
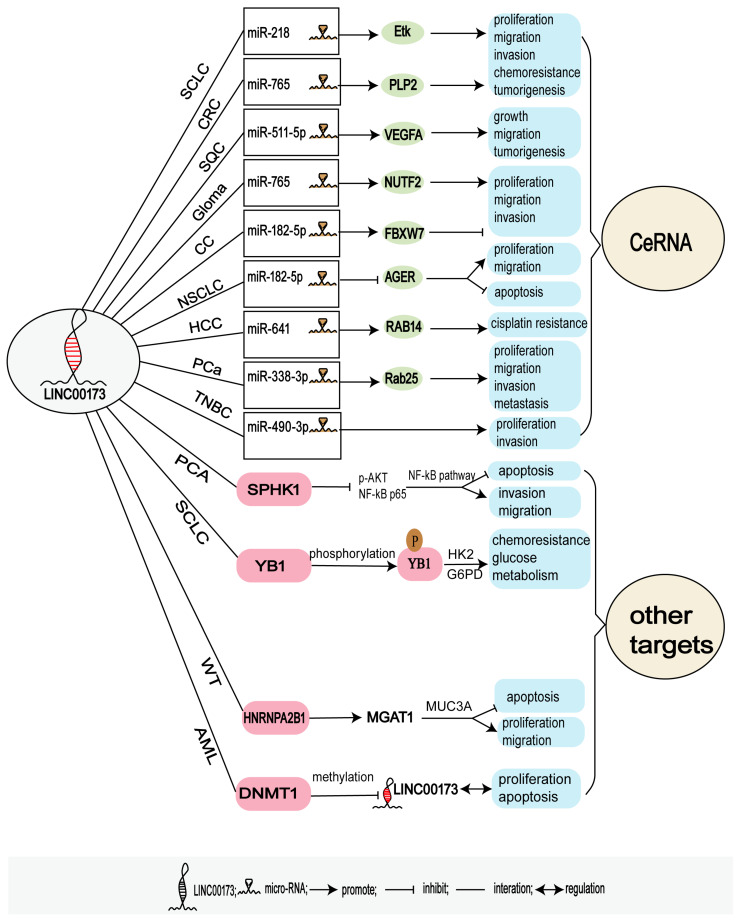
The regulatory targets related to LINC00173. LINC00173 interacts with 8 ceRNAs in 9 cancers and 4 other targets in 4 cancers.

**Table 1 cancers-14-05923-t001:** LINC00173 is dysregulated in multiple cancers and its functions.

Cancer	Dysregulated	Tumorigenesis	Subcellular Localization	Key Factors	Function	Pmid
NSCLC	Downregulated	Tumor suppressor	/	miR-182-5p, AGER, NF-kB	Promote proliferation, migration, inhibit apoptosis	31396332
	Upregulated	/	/	/	Potential diagnosis biomarker	32623390
SQC	Upregulated	Oncogenic	Cytoplasm	miR-511-5p, ΔNp63	Promote angiogenesis and tumorigenesis	32473645
SCLC	Upregulated	Oncogenic	Nucleus and Cytoplasm	YB1, HK2, G6PD, PPP	Promote chemoresistance and regulate glucose metabolism	34763086
	Upregulated	Oncogenic	Nucleus and Cytoplasm	miR-218, Etk, β-catenin GSKIP, NDRG1	Promote proliferation, migration, invasion, chemoresistance, and tumorigenesis	31477834
HCC	Upregulated	Oncogenic	/	miR-641, RAB14	Enhance CDDP resistance	33777195
CRC	Upregulated	Oncogenic	Cytoplasm	miR-765, PLP2	Promote growth, migration, invasiveness, and chemoresistance	32494200
ESCC	Upregulated	Oncogenic	/	/	Promote proliferation and cell cycle	29891973
Glioma	Upregulated	Oncogenic	/	miR-765, NUTF2	Promote proliferation, migration, and invasion	34603387
TNBC	Upregulated	Oncogenic	/	miR-490-3p	Promote proliferation and invasion	33141309
PCa	Upregulated	Oncogenic	Cytoplasm	miR-338-3p, Rab25	Promote proliferation, migration, invasion, and metastasis	33015770
WT	Upregulated	Oncogenic	/	MGAT1, MUC3A, HNRNPA2B1	Promote invasion and migration, inhibit apoptosis	35491865
PCA	Upregulated	Tumor suppressor	/	SPHK1, AKT, NF-kB	Suppress proliferation, invasion, and tumor growth, promote apoptosis	33978457
CC	Downregulated	Tumor suppressor	Nucleus and Cytoplasm	miR-182-5p, FBXW7	Repress proliferation, invasiveness, and migration	34407056
AML	Downregulated	Tumor suppressor	/	DNMT1	Enhance proliferation and tumor growth, inhibit apoptosis and chemoresistance to the CDDP	35135009

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
