# Peer review of "Long Intergenic Non-Protein Coding RNA 173 in Human Cancers"

_cancers, 2022, doi:10.3390/cancers14235923_

Round 1

Reviewer 1 Report

1. Line 32: mRNA by definition is not a ncRNA.

2. Table 1: Define table initials. Cervical Cancer (CC) apparently contradictory its function, since it is not oncogenic.

3. Lines 190-191: erratum, is talking about CC and not CRC. Here you would talk about the protective function of LINC00173 against cancer, mentioned by me in the previous point.

4. Line 278: FBXW7 is not an oncogene, but on the contrary, it is a tumor suppressor gen.

5. Line 324: In the caption of figure 2 there is an expression error "including the and the.

6. Figure 1: error, in the middle of the figure on the left, you should put "cisplatin sensitivity".

Reviewer 2 Report

The authors in the review “Long Intergenic Non-Protein Coding RNA 173 in Human 2 Cancers” have elaboratively described the role of the same in various perspectives especially its pro and anti-oncogenic involvement. They have collected most of the information regarding LINC00173 and its involvement indifferent types of cancers with interactions with DNA methylations, miRNA, CeRNAs and other targets. This review is up to the mark for publication with some minor comments/additions.

1)    Intermixed use of abbreviations and full should be avoided, and proper flow of its use should be maintained in the paper. For example in sub-sections of  2.1, 2.2 etc abbreviations are used for each main disease and inside the paragraphs full forms are again used (line 194 – “Acute myeloid leukemia” is used instead of AML).

2)    Heading can be full form of the abbreviations in subsections of 2.1, 2.2 etc

Thank you.
